# Annual and durable HIV retention in care and viral suppression among patients of Peter Ho Clinic, 2013-2017

**Debbie Y. Mohammed**[1,2]*, **Lisa Marie Koumoulos**[1,3], **Eugene Martin**[4], **Jihad Slim**[2,5]

**1** Department of Nursing, William Paterson University, Wayne, New Jersey, United States of America, **2** Division of Infectious Diseases, Saint Michael's Medical Center, Newark, New Jersey, United States of America, **3** Department of Quality, Palisades Medical Center, Hackensack Meridian Health, North Bergen, New Jersey, United States of America, **4** Department of Pathology and Laboratory Medicine, Rutgers-Robert Wood Johnson Medical School, Somerset, New Jersey, United States of America, **5** New York Medical College, Valhalla, New York, United States of America

* Mohammedd1@wpunj.edu

## Abstract

### Objectives

To determine rates of annual and durable retention in medical care and viral suppression among patients enrolled in the Peter Ho Clinic, from 2013–2017.

### Methods

This is a retrospective review of medical record data in an urban clinic, located in Newark, New Jersey, a high prevalence area of persons living with HIV. Viral load data were electronically downloaded, in rolling 1-year intervals, in two-month increments, from January 1, 2013 to December 31, 2019. Three teams were established, and every two months, they were provided with an updated list of patients with virologic failure. Retention and viral suppression rates were first calculated for each calendar-year. After patients were determined to be retained/suppressed annually, the proportion of patients with durable retention and viral suppression were calculated in two, three, four, five and six-year periods. Descriptive statistics were used to summarize sample characteristics by retention in care, virologic failure and viral suppression with Pearson Chi-square; p-value <0.05 was statistically significant. Multiple logistic regression models identified patient characteristics associated with retention in medical care, virologic failure and suppression.

### Results

As of December 31, 2017, 1000 (57%) patients were retained in medical care of whom 870 (87%) were suppressed. Between 2013 and 2016, decreases in annual (85% to 77%) and durable retention in care were noted: two-year (72% to 70%) and three-year (63% to 59%) periods. However, increases were noted for 2017, in annual (89%) and durable retention in the two-year period (79%). In the adjusted model, when compared to current patients, retention in care was less likely among patients reengaging in medical care (adjusted Odds Ratio

**Data Availability Statement:** All relevant data are within the manuscript

**Funding:** The authors received no specific funding for this work.

**Competing interests:** The authors have declared no competing interests.

(aOR): 0.77, 95% CI: 0.61–0.98) but more likely among those newly diagnosed from 2014–2017 (aOR: 1.57, 95% CI: 1.08–2.29), compared to those in care since 2013. A higher proportion of patients re-engaging in medical care had virologic failure than current patients (56% vs. 47%, p < 0.0001). As age decreased, virologic failure was more likely (p<0.0001). Between 2013 and 2017, increases in annual (74% to 87%) and durable viral suppression were noted: two-year (59% to 73%) and three-year (49% to 58%) periods. Viral suppression was more likely among patients retained in medical care up to 2017 versus those who were not (aOR: 5.52, 95% CI: 4.08–7.46). Those less likely to be suppressed were 20–29 vs. 60 years or older (aOR: 0.52, 95% CI: 0.28–0.97), had public vs. private insurance (aOR: 0.29, 95% CI: 0.15–0.55) and public vs. private housing (aOR: 0.59, 95% CI: 0.40–0.87).

## Conclusions

Restructuring clinical services at this urban clinic was associated with improved viral suppression. However, concurrent interventions to ensure retention in medical care were not implemented. Both retention in care and viral suppression interventions should be implemented in tandem to achieve an end to the epidemic. Retention in care and viral suppression should be measured longitudinally, instead of cross-sectional yearly evaluations, to capture dynamic changes in these indicators.

## Introduction

An estimated 1.04 million persons were living with HIV (PLWH) in the United States (U.S.) in 2018, with a prevalence of 374.6 per 100,000 population [1]. Males accounted for 75%, of whom 35% were Black, 73% were males who had sex with males (MSM), and injection drug use (IDU) was 9%. Among females, 58% were Black of whom 77% reported heterosexual contact or IDU (20%) as their transmission risk. In comparison, prevalence rates were higher in New Jersey and Essex County, 419.7 and 1,194.9 per 100,000 population, respectively [2]. In Essex County, characteristics of PLWH in this urban area reflect differences when compared to national data. Males accounted for a lower proportion of PLWH (61%), of whom a higher proportion were Black (69%), a lower proportion were MSM (35%) and a higher proportion reported IDU (16%) [2]. Among females, a higher proportion were Black (81%), of whom a lower proportion reported heterosexual contact (67%) [2]. The distribution of PLWH in Newark was similar to Essex County [3]. The Peter Ho Clinic (PHC) located in the City of Newark, is in the County of Essex.

Retention in medical care and viral suppression are beneficial to both the individual and wider community. For PLWH, mortality and opportunistic events decrease, even in patients with very advanced infection, with a concurrent increase in life expectancy [4–6]. Among PLWH with suppressed viral loads, ongoing transmission of infection decreases dramatically [7]. Despite these benefits, there are challenges to retention in medical care and viral suppression. In 2013, at the national level, 71% of PLWH were retained in care, of whom 77% were suppressed [8]. In comparison at the PHC, of 1,229 PLWH in medical care in 2013, 85% were retained for one year of whom 74% were suppressed.

Age, insurance, income, housing, and drug use were previously reported to impact retention in care and viral suppression. In 2013, the highest proportion of PLWH, retained in medical care were aged 45–54 (72.3%) versus (vs.) those 25–34 years old (69.8%) and viral

suppression increased with older age (43.7% and 57.5%, 25–34 and 45–54 years old, respectively) [8]. In a survey conducted in a public health HIV care relinkage program, 124 (50%) PLWH reported that not having insurance was a barrier to retention in care [9]. In another study, among women who did not take part in the AIDS Drug Assistance Program, those with Medicaid or no insurance were more likely to experience virologic failure compared to those with private insurance [10]. In addition, poverty and lack of health insurance were noted to be predictors of mortality among Black, heterosexual women [11]. A systemic review concluded that a lack of stable, secure, adequate housing was a significant barrier to consistent medical care, access, and adherence to anti-retroviral therapy (ART), sustained viral suppression, and risk of forward transmission, among PLWH [12]. Among those reporting IDU, retention in medical care and viral suppression results were equivocal. Sustained high viral loads were more likely among with a reported risk of IDU compared to heterosexuals [13]. In another evaluation, disparities in retention and viral suppression among PLWH with and without a history of IDU were eliminated by 2012, compared to 2000 [14].

A national strategy to combat gaps in care was introduced in 2010 and updated in 2015 [15, 16]. The National HIV/AIDS Strategy (NHAS) goals include improving health outcomes among PLWH. Measurable indicators include increasing the proportion of PLWH retained in medical care to at least 90% and viral suppression to at least 80%, by 2020. However, these indicators are based on cross-sectional data from yearly or 24-month periods, according to guidance from Health Resources and Services Administration, HIV/AIDS Bureau [17].

There is a paucity of reports on durable retention and viral suppression. One study conducted in Atlanta, Georgia reported retention in care for a two and three-year period (60% and 49%, respectively), from 2010 to 2013 [18]. In a national study conducted from 2011–2013, durable viral suppression, in a two-year period, was reported to be 61.8% [19]. However, researchers posit that true durability may best be defined after at least three years of viral suppression [20]. Durable retention over a five-year period was reported in two studies previously. In five sites of the HIV Research Network (HIVRN), five-year retention was 39.3%, while a clinical site in Lexington, Kentucky, reported 61% [21, 22]. Durable viral suppression from a clinical practice in Atlanta, Georgia, was 44% and 36%, in the two-year and three-year periods, respectively [18]. In a study evaluating the impact of care coordination in New York City, from 2009–2016, durable viral suppression was 37% in months 13–36 of follow-up [23]. In the U.S. Military HIV Natural History Study, durable viral suppression at the five and ten-year periods were 85% and 54%, respectively [24, 25].

The results of this study, using cross-sectional and longitudinal data, will be used to assess progress of this clinic towards achieving the NHAS 2020 goals [15, 16]. In addition, factors that facilitate or serve as barriers to achieving these goals, will be identified to inform future interventions and provide valuable lessons to other clinics, serving urban populations with similar challenges.

## Methods

### Design

This is a retrospective review of medical records data evaluating annual and durable retention in medical care and viral suppression, among patients enrolled at PHC, from 2013–2017.

### Setting

The PHC is located on the campus of Saint Michael's Medical Center, an urban academic institution in Newark, New Jersey. Newark is the epicenter of the epidemic in New Jersey and part of the New York metropolitan area that includes New York City. This is the first clinic in the

state to provide medical care for PLWH and serves approximately 1,200 persons yearly. Co-located services include HIV testing, access to pre-exposure prophylaxis and linkage to care coordinators. Clinical staff include infectious diseases providers and fellows, nurse practitioners, medical and non-medical case-managers, and a phlebotomist. Specialty co-located practices include gynecology, medication assisted therapy for opioid use and pain management services.

## Study population

Patients included in this study were at least 18 years old and alive as of December 31$^{st}$ of the respective year. At least one time in 2013–2017, they saw a medical provider, received a prescription for ART, and had viral load results documented in the electronic medical record. They were included in the study at the time of their first medical visit *or* viral load in 2013–2017 to six months after the last documented viral load or medical visit. Six patients were excluded from the virologic failure analysis as their viral loads were less than 200 copies/mL at the time of diagnosis (Fig 1). The medical records for PLWH not retained in medical care were evaluated for documentation of moving to another facility or death; if there was no documentation, they were considered lost to care.

## Viral suppression intervention

Beginning in 2014, clinical services were restructured for patients with viral loads of at least 200 copies/mL. Three teams were established, each composed of a clinician (either a nurse practitioner or doctor), a case manager, a nurse, and operational staff. Viral load data were

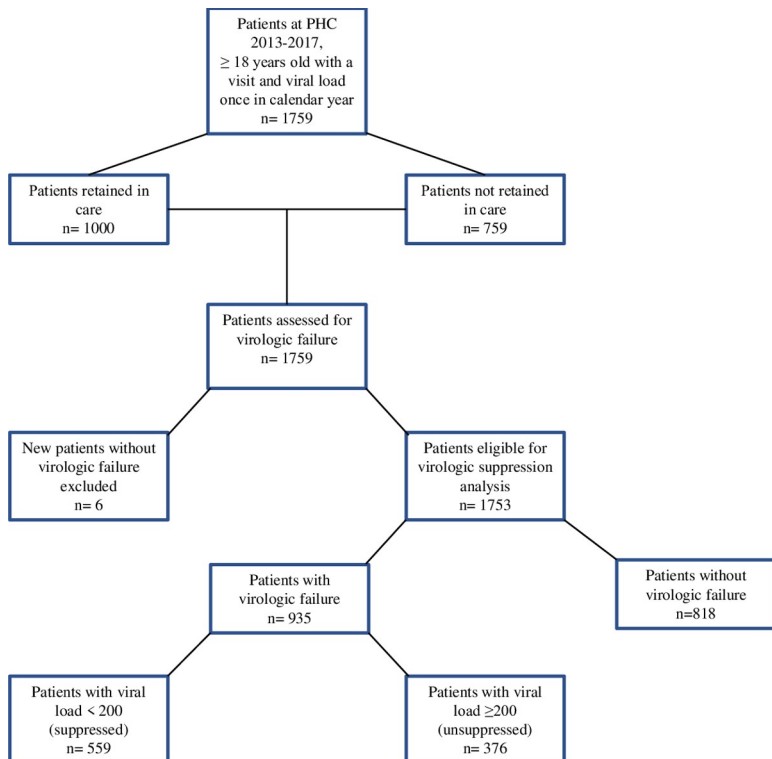

**Fig 1. Flowchart of patients assessed for retention in care, virologic failure and viral suppression, Peter Ho Clinic, 2013–2017.**

electronically downloaded, in rolling 1-year intervals, in two-month increments, from January 1, 2014 to December 31, 2019. Every two months, each team was provided with an updated list of patients with virologic failure. Initially, staff met weekly to discuss challenges associated with complex patients, and as their confidence increased, meetings were decreased to every two weeks and then monthly intervals.

A nurse practitioner reviewed genotype results and the electronic medical record to ensure that patients were on appropriate ART. Case managers and nurses assessed barriers to adherence and medical care and implemented appropriate interventions. Patients with difficulty adhering to ART came in weekly for prefilled medication boxes and counseling. They were transitioned to prefilled pharmacy packages once they became suppressed. The registration staff were key in scheduling and following up with phone reminders for appointments. Substance use and mental health counseling and treatment services were provided on-site. Referrals were also made to off-site locations, as indicated; those with unstable housing were referred to the New Jersey HIV Housing Collaborative for assistance.

## Outcomes

The primary outcomes for this study included the proportion of patients: a) retained in medical care annually, b) retained in durable medical care, c) with viral suppression annually, and d) durable viral suppression. Annual retention in medical care included documentation of at least two medical encounters, at least 90 days apart, at PHC in the respective calendar-year (*Yes*, *No*). A medical encounter included a medical visit with a provider with prescribing privileges *or* an HIV viral load test based on the core performance indicator, released in 2019 [17]. Additionally, patients received prescriptions for ART. Data were collected up to December 31st, 2019 to evaluate durable retention and viral suppression. Durable retention in care was defined as having at least two medical encounters and ART, in *each* calendar-year at least 90 days apart, over a two, three, four, five or six-year period. After patients were determined to be retained in care annually, the proportion of patients retained in care for each time-period was calculated [19]. Virologic failure was defined as a viral load of at least 200 copies/mL, at any time (*Yes*, *No)* [26]. Annual viral suppression was defined as all viral loads less than 200 copies/mL, in the respective calendar-year (*Yes*, *No*). Durable viral suppression was defined as all viral load values less than 200 copies/mL over a two, three, four, five or six-year period [19]. After the viral suppression status, for each patient, was determined by each year, the proportion who had durable viral suppression in each time-period was calculated. Secondary outcomes included: predictors of retention in medical care, virologic failure and viral suppression.

## Covariates

Patient's status at the time of the study was defined as Current, New and Reengaged. Current patients were those in care in 2013. In the respective calendar-year, new patients were newly diagnosed and those reengaged did not previously receive care at PHC. Demographic variables included age (*18–29, 30–49, 50–59 and ≥ 60 years*), sex at birth (*male, female*), race/ethnicity (*non-Hispanic Black, Hispanic, Other [non-Hispanic White, Asian, American Indian/Alaskan Native, Native Hawaiian, and multiple races*), transmission risk was based on the Centers for Disease Control and Prevention hierarchy (*male-to-male sex [MSM], IDU, heterosexual*) [27], insurance (*Medicaid, Medicare, private, none* [the medical care of these patients were paid by charity care, grant funding from the state or Ryan White HIV/AIDS Program]), housing (*private, public*), income based on the federal poverty limit (FPL) for one person in 2017, (>*FPL, ≤FPL*) [28], history of drug use (*heroin, cocaine, both, none*) and mental illness. Mental illness

(*Yes/No*) was a composite of patient self-report, routine psychiatric visits, and medications documented in the electronic medical record.

In the regression analyses, categories of the following covariates were collapsed where outcomes were similar: age (*<60, ≥ 60 years*), race (*non-Hispanic Black or Hispanic, Other*), insurance (*Public [included Medicaid, Medicare, None], Private*) and drug use (*Yes, No*).

## Data analysis

This study was approved by the Saint Michael's Medical Center Institutional Review Board. Data collection were completed as of December 31st, 2019. The Statistical Analysis System (SAS, version 9.4) Cary, North Carolina, was used for all analyses. After quality checks, the data were stripped of personal identifiers prior to analyses. Descriptive statistics were used to summarize sample characteristics by retention in medical care, virologic failure and viral suppression with Pearson's Chi-square. A p-value <0.05 was considered statistically significant. Multiple logistic regression models, using backwards deletion, were developed to identify factors associated with retention in medical care in 2017; virologic failure, 2013–2017; and viral suppression, 2013–2017, for those who experienced virologic failure. The selected variables were based on clinical and epidemiological significance. Collinearity in the category for risk transmission was mitigated by adhering to the hierarchical categorization developed by the Centers for Disease Control and Prevention [27], and categorizing drug use as cocaine only, heroin only and those with a history of using both drugs. Model fit was evaluated using Hosmer and Lemeshow Goodness-of-Fit test.

## Results

From 2013 to 2017, a total of 1,759 PLWH received medical care at the PHC (Table 1). At least 60% were 50 years or older, male (64%) or Black (69%). From 2013 to 2017, shifts in the proportion of some groups were noted.

By patients' status, this included 1229 (70%) current patients, 157 (9%) newly diagnosed and 373 (21%) who reengaged in medical care (Table 2). Characteristics were similar for current patients and those reengaging in medical care.

However, when new patients were compared with those reengaging in care, higher proportions were < 60 years old (92% vs. 83%), male (82% vs. 67%), reported MSM (52% vs. 30%), had private insurance (23% vs. 14%), private housing (96% vs. 80%), or income > FPL (42% vs. 18%), respectively, p<0.0001. Higher proportions of patients reengaging in medical care were Black or Hispanic (91% vs. 87%), reported a history of drug use (42% vs. 17%) and mental illness (35% vs. 13%) compared to new patients, (p<0.0001).

Annual retention in medical care decreased from 2013 to 2016, 85% to 77%, respectively (Fig 2). Higher proportions of patients were retained in the two (82% vs. 72%), three (66% vs. 63%) and four-year (61% vs. 53%) periods for patients from 2015, compared to 2013. In 2017, the proportions for the annual and two-year period were 89% and 79%, respectively.

The proportion of patients moving declined from 2013 to 2017, 9% to 2% respectively, however, those lost to care, was almost two times higher in 2016 compared to 2015 (Fig 3).

Overall, 1,000 patients (57%) were alive and retained in medical care as of December 31, 2017 (Table 3). Retention in medical care was higher for newly diagnosed patients (70%) compared to those in care since 2013 (51%) or reengaging in medical care (49%), p<0.0001 (Table 3 and Fig 4).

Factors associated with retention in medical care in the adjusted model included status, age, risk, income, and drug use. When compared to current patients, retention in care was less likely among patients reengaging in medical care (aOR: 0.77, 95% CI: 0.61–0.98) but more

**Table 1. Characteristics of patients, Peter Ho Clinic, New Jersey (2013–2017).**

| Characteristic | 2013 | 2014 | 2015 | 2016 | 2017 | Total |
|---|---|---|---|---|---|---|
| | N (%) | N (%) | N (%) | N (%) | N (%) | N (%) |
| **Age** | | | | | | |
| 18–29 | 40 (3) | 55 (5) | 72 (6) | 82 (7) | 75 (7) | 140 (8) |
| 30–39 | 103 (9) | 116 (10) | 130 (11) | 127 (11) | 112 (11) | 215 (12) |
| 40–49 | 224 (18) | 230 (19) | 228 (19) | 204 (18) | 181 (18) | 319 (18) |
| 50–59 | 469 (38) | 447 (36) | 428 (36) | 408 (36) | 347 (35) | 617 (35) |
| ≥60 | 393 (32) | 372 (30) | 344 (28) | 319 (28) | 285 (29) | 468 (27) |
| **Gender** | | | | | | |
| Female | 478 (39) | 454 (37) | 451 (38) | 413 (36) | 362 (36) | 631 (36) |
| Male | 751 (61) | 766 (63) | 751 (62) | 727 (64) | 638 (64) | 1128 (64) |
| **Race/ethnicity** | | | | | | |
| Black Non-Hispanic | 868 (70) | 862 (71) | 846 (70) | 802 (70) | 700 (70) | 1222 (69) |
| Hispanic | 278 (23) | 279 (23) | 279 (23) | 267 (24) | 231 (23) | 400 (23) |
| Other[a] | 83 (7) | 79 (6) | 77 (7) | 71 (6) | 69 (7) | 137 (8) |
| **Transmission Risk** | | | | | | |
| Male-to-Male sex | 218 (18) | 251 (20) | 267 (22) | 266 (23) | 232 (23) | 413 (23) |
| Injection Drug Use | 242 (20) | 215 (18) | 193 (16) | 171 (15) | 137 (14) | 295 (17) |
| Heterosexual sex | 769 (62) | 754 (62) | 742 (62) | 703 (62) | 631 (63) | 1051 (60) |
| **Insurance** | | | | | | |
| Medicaid | 639 (52) | 633 (52) | 610 (51) | 571 (50) | 486 (48) | 897 (51) |
| Medicare | 220 (18) | 218 (18) | 200 (17) | 195 (17) | 169 (17) | 267 (15) |
| Private | 136 (11) | 150 (12) | 169 (14) | 168 (15) | 148 (15) | 224 (13) |
| None[b] | 234 (19) | 219 (18) | 223 (18) | 206 (18) | 197 (20) | 371 (21) |
| **Housing** | | | | | | |
| Private | 995 (81) | 1012 (83) | 1011 (84) | 965 (85) | 848 (85) | 1444 (82) |
| Public | 234 (19) | 208 (17) | 191 (16) | 175 (15) | 152 (15) | 315 (18) |
| **Income** | | | | | | |
| >FPL[c] | 355 (29) | 375 (31) | 388 (32) | 382 (33) | 340 (34) | 526 (30) |
| ≤FPL[c] | 874 (71) | 845 (69) | 814 (68) | 758 (67) | 660 (66) | 1233 (70) |
| **Drug Use** | | | | | | |
| Heroin | 70 (6) | 60 (5) | 52 (4) | 51 (5) | 32 (3) | 91 (5) |
| Cocaine | 154 (12) | 153 (13) | 142 (12) | 137 (12) | 116 (12) | 221 (13) |
| Both | 341 (28) | 330 (27) | 287 (24) | 252 (22) | 214 (21) | 437 (25) |
| None | 664 (54) | 677 (55) | 721 (60) | 700 (61) | 638 (64) | 1011 (57) |
| **Mental Illness** | | | | | | |
| Yes | 481 (39) | 450 (37) | 408 (34) | 381 (34) | 325 (33) | 630 (36) |
| No | 748 (61) | 770 (63) | 794 (66) | 759 (66) | 675 (67) | 1129 (64) |
| **Total** | 1229 | 1220 | 1202 | 1140 | 1000 | 1759 |

[a] Includes Non-Hispanic white, Asian, American Indian, Alaskan Native, Native Hawaiian, and multiple races

[b] Patient care was paid for by charity care, state funding, or Ryan White HIV/AIDS program

[c] FPL- Federal Poverty Level for 2017

likely among those who were newly diagnosed from 2014–2017 (aOR:1.57, 95% CI: 1.08–2.29). Those less likely to be retained in medical care were younger than 60 vs. > 60 years old (aOR: 0.71, 95% CI: 0.56–0.90), reported IDU vs. heterosexual contact (aOR: 0.71, 95% CI: 0.53–0.95), had an income < FPL vs. > FPL for 2017 (aOR: 0.75, 95% CI: 0.59–0.95) and reported a history of drug use vs. none (aOR: 0.64, 95% CI: 0.51–0.80).

**Table 2. Characteristics of patients by status, Peter Ho Clinic, New Jersey (2013–2017).**

| Characteristics | Reengaged | New | Current | Total |
|---|---|---|---|---|
| | N (%) | N (%) | N (%) | 1759 |
| | 373 (21) | 157 (9) | 1229 (70) | |
| **Age** | | | | p<0.0001 |
| < 60 | 309 (83) | 145 (92) | 837 (68) | 1291 (73) |
| ≥ 60 | 64 (17) | 12 (8) | 392 (32) | 468 (26) |
| **Gender** | | | | p<0.0001 |
| Female | 126 (33) | 29 (18) | 478 (39) | 631 (36) |
| Male | 247 (67) | 128 (82) | 751 (61) | 1126 (64) |
| **Race/ethnicity** | | | | p<0.0001 |
| Black or Hispanic | 339 (91) | 138 (87) | 1147 (93) | 1624 (92) |
| Other[a] | 34 (9) | 19 (13) | 82 (7) | 137 (8) |
| **Transmission Risk** | | | | p<0.0001 |
| Male-to-Male sex | 112 (30) | 82 (52) | 219 (18) | 413 (23) |
| Injection Drug Use | 49 (13) | 4 (2) | 242 (20) | 295 (17) |
| Heterosexual sex | 210 (57) | 72 (46) | 768 (62) | 1051 (60) |
| **Insurance** | | | | p<0.0001 |
| Public[b] | 320 (86) | 122 (77) | 1093 (89) | 1535 (87) |
| Private | 53 (14) | 35 (23) | 136 (11) | 224 (13) |
| **Housing** | | | | p<0.0001 |
| Public | 74 (20) | 7 (4) | 234 (19) | 315 (18) |
| Private | 299 (80) | 150 (96) | 995 (81) | 1444 (82) |
| **Income** | | | | p<0.0001 |
| ≤FPL[c] | 266 (72) | 92 (58) | 875 (71) | 1233 (70) |
| >FPL[c] | 107 (18) | 65 (42) | 354 (29) | 526 (30) |
| **Drug Use** | | | | p<0.0001 |
| Yes[d] | 156 (42) | 27 (17) | 565 (46) | 748 (43) |
| No | 217 (58) | 130 (83) | 664 (54) | 1011 (57) |
| **Mental Illness** | | | | p<0.0001 |
| Yes | 128 (35) | 21 (13) | 481 (39) | 630 (36) |
| No | 245 (65) | 136 (87) | 748 (61) | 1129 (64) |

[a]Other–includes Non-Hispanic White, Asian, American Indian/Alaskan Native, Native Hawaiian, multiple races

[b]Public insurance include Medicare, Medicaid, and No insurance.

[c]FPL–Federal Poverty Limit for 2017

[d]Drug Use (Yes) included heroin and cocaine

A viral load of at least 200 copies/mL was present at least one time for 935 (53%) patients, from 2013 to 2017 (Table 4). A higher proportion of patients reengaging in medical care had virologic failure than those in care since 2013, (56% vs. 46%, p < 0.0001) (Fig 4). In the adjusted model age, insurance, income, and drug use were associated with virologic failure.

Virologic failure was more likely among those 20–29 (aOR 7.70: 95% CI: 4.70–12.64), 30–39 (aOR: 3.38: 95% CI: 2.32–4.92), 40–49 (aOR: 2.44: 95% CI: 1.79–3.32), and 50–59 (aOR:1.36, 95% CI: 1.05–1.75) vs. >60 years old; with public vs. private insurance (aOR: 1.62, 95% CI: 1.17–2.26), income at < FPL vs. > FPL for 2017 (aOR:1.35, 95% CI: 1.06–1.71) and a history of drug use vs. none (aOR:1.38, 95% CI: 1.09–1.76).

Annual viral suppression increased over time: 2013 (74%), 2014 (77%), 2015 (81%), 2016 (86%) and 2017 (87%) (Fig 5). Higher proportions of patients in care since 2013 were virally

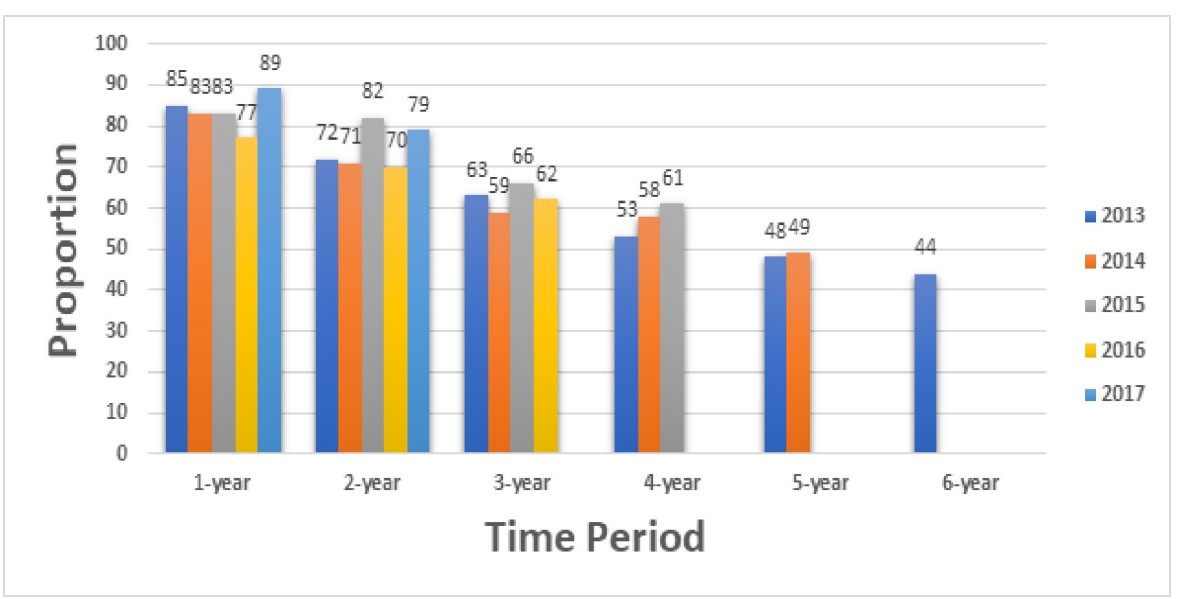

**Fig 2. Retention in medical care and durable retention in medical care, by time-period.**

suppressed, compared to patients reengaging in care (64% vs. 54%, respectively, p<0.0001). Durable viral suppression increased among patients in the two (59% to 73%) and three-year (49% to 58%) periods, from 2013 to 2017. Similar proportions were noted for the four to six-year periods.

Among patients with virologic failure, from 2013–2017, 559 (59%) were suppressed at the last reported result (Table 5). They included new patients (76%), those who were retained in care in 2017 (77%) or were at least 60 years old (66%). In the adjusted model, status, retention in medical care in 2017, age, insurance and housing were associated with viral suppression.

Viral suppression was more likely among new patients vs. current patients (aOR:1.86, 95% CI:1.10–3.14) and patients retained in medical care in 2017 vs. those who were not (aOR: 5.32: 95% CI: 3.93–7.20). Viral suppression was less likely in patients 20–29 vs. >60 years old (aOR: 0.41, 95% CI:0.21–0.81), with public vs. private insurance (aOR: 0.29, 95% CI: 0.15–0.57), or who lived in public vs. private housing (aOR: 0.62, 95% CI: 0.42–0.92).

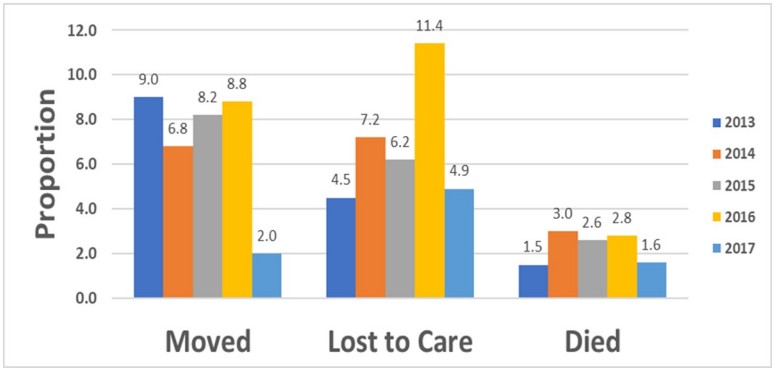

**Fig 3. Patients moved, lost to care or died, by year, 2013–2017.**

**Table 3. Factors associated with retention in care, Peter Ho Clinic, 2013–2017.**

| Characteristic | Retained in Care | | P value | Odds Ratio 95% CI[a] | Adjusted Odds Ratio 95% CI[a] |
|---|---|---|---|---|---|
| | Yes | No | | | |
| | N (%) | N (%) | | | |
| | 1000 (57) | 759 (43) | | | |
| **Status** | | | <0.0001 | | |
| New | 110 (70) | 47 (30) | | 1.71 (1.19–2.44) | 1.57 (1.08–2.29) |
| Reengaged | 185 (49) | 188 (51) | | 0.74 (0.59–0.94) | 0.77 (0.61–0.98) |
| Current | 705 (57) | 524 (43) | | 1.00 | 1.00 |
| **Age** | | | 0.0391 | | |
| <60 | 715 (55) | 576 (45) | | 0.80 (0.64–0.99) | 0.71 (0.56–0.90) |
| ≥60 | 285 (61) | 183 (39) | | 1.00 | 1.00 |
| **Gender** | | | 0.7425 | | |
| Female | 362 (57) | 269 (43) | | 1.03 (0.85–1.26) | 1.00 (0.79–1.26) |
| Male | 638 (56) | 490 (43) | | 1.00 | 1.00 |
| **Race/ethnicity** | | | 0.1105 | | |
| Black or Hispanic | 931 (57) | 691 (43) | | 1.33 (0.94–1.88) | 1.34 (0.93–1.92) |
| Other[b] | 69 (50) | 68 (50) | | 1.00 | 1.00 |
| **Transmission Risk** | | | 0.0002 | | |
| Male-to-Male sex | 232 (56) | 181 (44) | | 0.85 (0.68–1.07) | 0.77 (0.58–1.02) |
| Injection Drug Use | 137 (46) | 158 (54) | | 0.58 (0.45–0.75) | 0.71 (0.53–0.95) |
| Heterosexual sex | 631 (60) | 420 (40) | | 1.00 | 1.00 |
| **Insurance** | | | 0.0029 | | |
| Public | 852 (56) | 683 (44) | | 0.64 (0.48–0.86) | 0.85 (0.61–1.19) |
| Private | 148 (66) | 76 (34) | | 1.00 | 1.00 |
| **Housing** | | | 0.0007 | | |
| Private | 848 (59) | 596 (41) | | 1.00 | 1.00 |
| Public[c] | 152 (48) | 163 (52) | | 0.66 (0.51–0.84) | 0.80 (0.62–1.03) |
| **Income** | | | <0.0001 | | |
| >FPL[d] | 340 (64) | 186 (36) | | 1.00 | 1.00 |
| ≤FPL[d] | 660 (54) | 573 (47) | | 0.63 (0.51–0.78) | 0.75 (0.59–0.95) |
| **Drug Use** | | | <0.0001 | | |
| Yes[e] | 362 (48) | 386 (52) | | 0.55 (0.45–0.66) | 0.64 (0.51–0.80) |
| No | 638 (63) | 373 (37) | | 1.00 | 1.00 |
| **Mental Illness** | | | 0.0009 | | |
| Yes | 325 (52) | 305 (48) | | 0.72 (0.59–0.87) | 1.01 (0.81–1.26) |
| No | 675 (60) | 454 (40) | | 1.00 | 1.00 |

[a]CI–Confidence Interval

[b]Other–includes Non-Hispanic White, Asian, American Indian/Alaskan Native, Native Hawaiian, multiple races

[c] Public insurance include Medicare, Medicaid, and No insurance.

[d]FPL–Federal Poverty Limit for 2017

[e] Drug Use (Yes) included heroin and cocaine

## Discussion

Annual viral suppression at PHC increased by 18% from 2013 to 2017. This increase coincides with restructuring of clinic services for patients with a viral load of at least 200 copies/mL. Despite this attempt to improve viral suppression, a concurrent intervention to improve retention in care was not implemented and annual retention declined by 10% from 2013–2016.

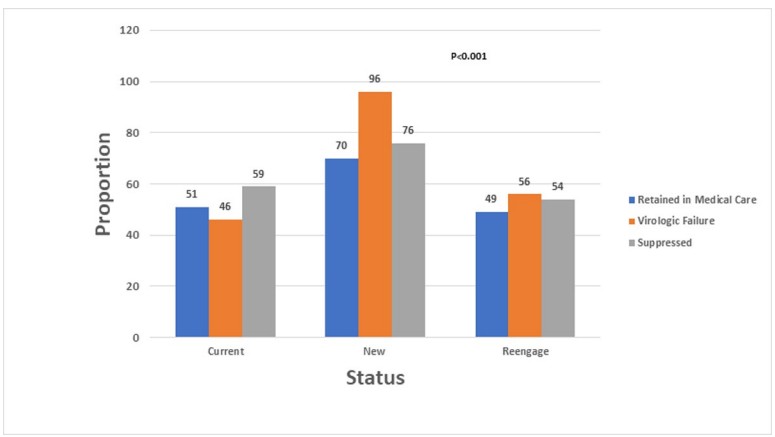

**Fig 4. Retention in medical care, virologic failure and suppression by status, Peter Ho Clinic, 2013–2017.** Note: Retention in medical care: Two medical encounters (medical visit or viral load), three months apart, in 2017, ART prescription and alive as of 12/31/2017. Virologic Failure: Most Recent Viral load > 200 copies/mL. Suppressed: Most Recent Viral load < 200 copies/mL. Status: Current patients were those in care in 2013. In the respective calendar-year, new patients were newly diagnosed and those reengaged did not previously receive care in PHC.

In this study, virologic failure did not vary by retention status. However, patients who were retained in care until 2017 were at least 5 times more likely to be suppressed, at the last reported result than those who were lost to care, moved, or died. This relationship between retention and viral suppression was previously highlighted. Among patients in a Minnesota clinic from 2003–2015, retention in care for every calendar-year was associated with sustained viral suppression [29]. Continuous retention in Atlanta, Georgia was associated with viral suppression at the end of 36 months (adjusted prevalence ratio 3.12; 95% CI, 2.40, 4.07) [18]. A study in Lexington, Kentucky from 2003–2011, followed patients an average of 6.2 years, and reported that individuals optimally retained in continuous care were almost 3 times more likely to maintain a suppressed viral load compared to those not retained (OR: 2.97; 95% CI: 1.65–5.32) [22]. Additionally, among PLWH who received medical care funded by the Ryan White HIV/AIDS Program in 2011, viral suppression was higher among retained clients (77.7%) vs. clients who were not retained (58.3%) [30].

In this study, we used the recently published definition of annual retention in medical care of two medical encounters (visit with a medical provider *or* viral load results), 90 days apart, as it reflects a practical approach and is supported in previous studies [17]. The HIVRN reported that 10% of patients from eleven HIV care sites who did not meet the definition for retention, completing 2 or more HIV medical visits separated by ≥90 days apart in a calendar year, remained virally suppressed [21]. In another study, the HIVRN combined data from patients who were eligible for Medicaid and pharmacy utilization in four clinical sites [31]. The researchers noted that patients who continued to receive ART during gaps in care had a suppressed viral load closest to the gap. These findings provide support for less frequent medical visits/viral loads among patients with viral suppression compared to those with virologic failure.

In this study we did cross-sectional as well as longitudinal evaluations of retention in medical care and viral suppression. Recent studies highlight the importance of measuring retention and viral suppression in a longitudinal manner as this better reflects the true status of care over time [29, 32–34]. Dynamic viral load trajectories are easily overlooked with cross-sectional evaluation of the last value while longitudinal measures provide more granular data [19]. This longitudinal evaluation may provide the information needed in planning future interventions at PHC, as well as other urban clinics.

**Table 4. Factors associated with virologic failure, Peter Ho Clinic, 2013–2017.**

| Characteristic | Virologic Failure | | p-value | Odds Ratio 95% CI[a] | Adjusted Odds Ratio 95% CI[a] |
|---|---|---|---|---|---|
| | **No** | **Yes** | | | |
| | N (%) | N (%) | | | |
| | 818 (47) | 935 (53) | | | |
| **Retained in Care** | | | 0.8237 | | |
| Yes | 468 (47) | 530 (53) | | 1.00 | 1.00 |
| No | 350 (47) | 405 (53) | | 1.02 (0.85–1.24) | 0.88 (0.72–1.08) |
| **Age** | | | <0.0001 | | |
| 20–29 | 28 (20) | 110 (80) | | 5.46 (3.47–8.60) | 7.70 (4.70–12.64) |
| 30–39 | 77 (36) | 137 (64) | | 2.47 (1.77–3.45) | 3.38 (2.32–4.92) |
| 40–49 | 128 (40) | 191 (60) | | 2.07 (1.55–2.77) | 2.44 (1.79–3.32) |
| 50–59 | 314 (51) | 302 (49) | | 1.34 (1.05–1.70) | 1.36 (1.05–1.75) |
| ≥60 | 271 (58) | 195 (42) | | 1.00 | 1.00 |
| **Gender** | | | 0.3435 | | |
| Female | 303 (48) | 326 (52) | | 0.91 (0.75–1.11) | 0.84 (0.67–1.06) |
| Male | 515 (46) | 609 (54) | | 1.00 | 1.00 |
| **Race/ethnicity** | | | 0.2072 | | |
| Black or Hispanic | 747 (46) | 869 (54) | | 1.25 (0.88–1.77) | 1.22 (0.84–1.76) |
| Other[b] | 71 (52) | 66 (48) | | 1.00 | 1.00 |
| **Transmission Risk** | | | 0.0953 | | |
| Male-to-Male sex | 177 (43) | 235 (57) | | 1.17 (0.93–1.47) | 0.75 (0.56–1.01) |
| Injection Drug Use | 150 (51) | 143 (49) | | 0.84 (0.65–1.09) | 0.84 (0.63–1.13) |
| Heterosexual sex | 491 (47) | 557 (53) | | 1.00 | 1.00 |
| **Insurance** | | | 0.0003 | | |
| Public[c] | 689 (45) | 841 (55) | | 1.68 (1.26–2.23) | 1.62 (1.17–2.26) |
| Private | 129 (58) | 94 (42) | | 1.00 | 1.00 |
| **Housing** | | | 0.8996 | | |
| Public | 148 (47) | 167 (53) | | 0.98 (0.77–1.27) | 0.95 (0.73–1.23) |
| Private | 670 (47) | 768 (53) | | 1.00 | 1.00 |
| **Income** | | | 0.0005 | | |
| ≤FPL[d] | 541 (44) | 690 (56) | | 1.44 (1.17–1.77) | 1.35 (1.06–1.71) |
| >FPL[d] | 277 (53) | 245 (47) | | 1.00 | 1.00 |
| **Drug Use** | | | 0.2210 | | |
| Yes[e] | 335 (45) | 410 (55) | | 1.13 (0.93–1.36) | 1.38 (1.09–1.76) |
| No | 483 (48) | 525 (52) | | 1.00 | 1.00 |
| **Mental Illness** | | | 0.0716 | | |
| Yes | 275 (44) | 353 (56) | | 1.20 (0.98–1.46) | 1.15 (0.91–1.44) |
| No | 543 (48) | 582 (51) | | 1.00 | 1.00 |

[a]CI–Confidence Interval

[b]Other–includes Non-Hispanic White, Asian, American Indian/Alaskan Native, Native Hawaiian, multiple races

[c] Public insurance include Medicare, Medicaid, and No insurance.

[d]FPL–Federal Poverty Limit for 2017

[e] Drug Use (Yes) included heroin and cocaine

Age was associated with retention and viral suppression in this evaluation and supported in previous reports. One national study reported a lower proportion of those 25–34 vs. > 55 years-old (52% and 72%, respectively) with durable (two-year) viral suppression [19]. A cross-sectional analysis, among fourteen cohorts, in the U.S. and Canada reported that (1) the older the

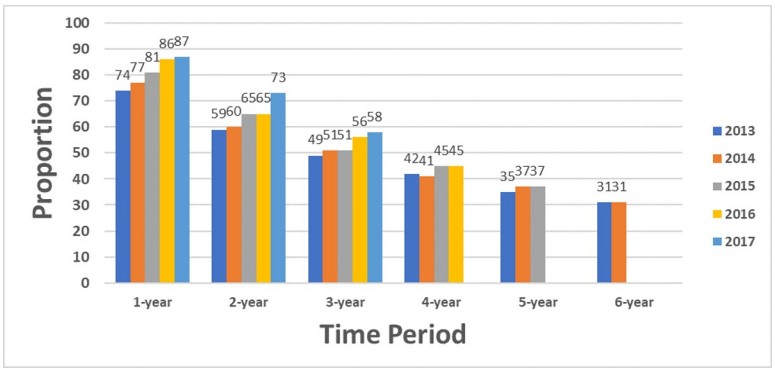

**Fig 5. Viral suppression and durable viral suppression by time period, 2013–2017.**

individual, the greater the probability of viral suppression in both the retained and not retained in care groups; (2) patients who were retained in care had a greater probability of viral suppression than those not retained in care and (3) the association between retention in care and viral suppression was greatest for younger versus older age groups [35]. Another study in the Southeastern U.S., compared those 18–24 and 35–44 years old and reported that younger patients were more likely to not be retained in medical care or have viral suppression [36].

In this study, patients who lived in public housing (including those with unstable housing) compared to private housing were 40% less likely to achieve viral suppression. This is supported in a recent national study among patients accessing care in Ryan White funded sites in 2018; those who lived in unstable compared to stable housing, had lower rates of viral suppression [37]. Previous reports indicate that public housing, when stable, mitigated barriers to viral suppression. A recent retrospective matched cohort study in New York City reported that low income PLWH who received housing services for more than one year, were more likely to be engaged in care and virally suppressed than similar populations who did not receive these services [38]. In another study, PLWH in continuous, stable housing in 2015, were reported to have continuous, stable viral suppression [39]. Conversely, time spent in emergency housing was a predictor of a lack of viral suppression. Identifying patients in unstable housing and transitioning them to stable housing will improve viral suppression rates, even if they live in public housing.

Insurance status was another structural barrier identified in this study. New Jersey expanded Medicaid and increased access to care for low-income residents, by implementing many reforms recommended by the Affordable Care Act starting in 2014 [40]. Despite the availability of public insurance, before being seen in PHC, there were many processes patients had to navigate to make this a reality. These included qualifying for hospital charity care if they did not have insurance [41], enrollment and reenrollment and requiring a primary care provider to issue referrals if they had a managed care plan. This may have contributed to patients not being retained in medical care or achieving viral suppression. Patients who are lost to care, more symptomatic, and/or racial/ethnic minorities may be less likely to overcome these barriers [42]. At least 70% of PLWH at PHC, qualified for federally funded services, based on an income less than the FPL for one person, in 2017, or had either Medicaid or no insurance (Charity Care/Ryan White funding), and were primarily racial and ethnic minorities (>90%). Numerous studies have documented the importance of Ryan White federal funding in maintaining PLWH in care and improved outcomes [30, 43–47]. Urban clinics will need to maximize the use of this resource to ensure that patients overcome structural barriers and achieve the goals of the NHAS.

**Table 5. Factors associated with viral suppression, among patients with virologic failure, Peter Ho Clinic, 2013–2017.**

| Characteristic | Suppressed | | p-value | Odds Ratio 95% CI [a] | Adjusted Odds Ratio 95% CI [a] |
|---|---|---|---|---|---|
| | Yes | No | | | |
| | N (%) | N (%) | | | |
| | 559 (59) | 376 (41) | | | |
| **Status** | | | <0.0001 | | |
| New | 115 (76) | 36 (24) | | 2.15 (1.43–3.22) | 1.86 (1.10–3.14) |
| Reengaged | 104 (50) | 104(50) | | 0.69 (0.50–0.95) | 0.77 (0.53–1.11) |
| Current | 340 (59) | 236 (41) | | 1.00 | 1.00 |
| **Retention in Medical Care** | | | <0.0001 | | |
| Yes | 408 (77) | 122 (23) | | 5.63 (4.23–7.48) | 5.32(3.93–7.20) |
| No | 151 (37) | 254 (63) | | 1.00 | 1.00 |
| **Age** | | | 0.3563 | | |
| 20–29 | 63 (57) | 47 (43) | | 0.69(0.42–1.11) | 0.41(0.21–0.81) |
| 30–39 | 82 (60) | 55 (40) | | 0.76 (0.49–1.20) | 0.59 (0.33–1.04) |
| 40–49 | 110 (58) | 81 (42) | | 0.70 (0.46–1.05) | 0.73 (0.45–1.19) |
| 50–59 | 175(58) | 127 (42) | | 0.71 (0.49–1.03) | 0.89 (0.58–1.36) |
| ≥60 | 129(66) | 66(34) | | 1.00 | 1.00 |
| **Gender** | | | 0.0053 | | |
| Female | 175 (54) | 151(46) | | 0.68(0.52–0.89) | 0.75(0.53–1.06) |
| Male | 384 (63) | 225(37) | | 1.00 | 1.00 |
| **Race/ethnicity** | | | 0.2370 | | |
| Black or Hispanic | 515 (59) | 354 (41) | | 0.73 (0.43–1.24) | 0.89(0.48–1.63) |
| Other[b] | 44(67) | 22(33) | | 1.00 | 1.00 |
| **Transmission Risk** | | | 0.3511 | | |
| Male-to-Male sex | 149 (63) | 86 (37) | | 1.26(0.92–1.72) | 1.13(0.73–1.76) |
| Injection Drug Use | 87 (61) | 56 (39) | | 1.13(0.77–1.64) | 1.52(0.96–2.41) |
| Heterosexual sex | 323 (58) | 234 (42) | | 1.00 | 1.00 |
| **Insurance** | | | <0.0001 | | |
| Public[c] | 479 (57) | 362 (43) | | 0.23(0.13–0.42) | 0.29(0.15–0.57) |
| Private | 80 (85) | 14 (15) | | 1.00 | 1.00 |
| **Housing** | | | <0.0001 | | |
| Public | 76 (45) | 91 (55) | | 0.48(0.35–0.69) | 0.62(0.42–0.92) |
| Private | 483 (63) | 285 (37) | | 1.00 | 1.00 |
| **Income** | | | <0.0001 | | |
| ≤FPL[d] | 382 (55) | 308 (45) | | 0.48(0.35–0.65) | 0.77(0.53–1.14) |
| >FPL[d] | 177 (72) | 68 (28) | | 1.00 | 1.00 |
| **Drug Use** | | | 0.0007 | | |
| Yes[e] | 220 (54) | 190 (46) | | 0.64(0.49–0.83) | 0.90(0.62–1.30) |
| No | 339 (65) | 186 (35) | | 1.00 | 1.00 |
| **Mental Illness** | | | 0.0024 | | |
| Yes | 189 (53) | 164 (47) | | 0.66(0.51–0.86) | 0.94(0.66–1.32) |
| No | 370 (64) | 212 (36) | | 1.00 | 1.00 |

[a]CI–Confidence Interval

[b]Other–includes Non-Hispanic White, Asian, American Indian/Alaskan Native, Native Hawaiian, multiple races

[c] Public insurance include Medicare, Medicaid, and No insurance.

[d]FPL–Federal Poverty Limit for 2017

[e] Drug Use (Yes) included heroin and cocaine

## Strengths and limitations

A strength of this study is that we examined retention in care and viral suppression independent of each other. The conditional cascade methodology requires success at upstream stages before measuring success at later stages and underreports performance by up to 20% compared with evaluating each stage separately [17, 48]. A second strength of this study is that we evaluated longitudinal measures of retention in care and viral suppression. This provides a broader, more comprehensive perspective of retention and viral suppression than cross-sectional evaluations alone [29, 33–34].

There are limitations to this study. This was a retrospective, observational study, so that we can only discuss associations and not causation. Secondly, this study may lack generalizability to other clinics as the results are applicable to a single, HIV specialty clinic in the Northeastern U.S. Thirdly, there may be other confounders that contributed to the reported associations. Fourthly, there were frequent changes in insurance type after the introduction of the Affordable Care Act that were not accounted for in this study, these likely impacted retention in care and viral suppression [40, 42]. Lastly, data from PHC were not matched to statewide or national HIV surveillance or vital statistics databases. We may have overestimated PLWH who were lost to care and underestimated those who were deceased.

In conclusion, we evaluated annual and durable retention in medical care and viral suppression using cross-sectional as well as longitudinal methods. The PHC is on-track towards achieving the NHAS goals by 2020 [16]. Concurrent interventions for retention in care and viral suppression should be implemented, with a focus on those reengaging in care. As we embark on ending the epidemic in the U.S., it will be useful to evaluate the NHAS in a longitudinal manner [17, 49]. Patients reengaging in medical care, younger than 60 years old, with public insurance or public housing may benefit from access to intensive care coordination to improve durable retention and viral suppression.

## Acknowledgments

**Patients**

Thank you to all the patients enrolled at Peter Ho Clinic, from 2013 to 2019, without whom this study would not be possible.

**Staff**

Thank you to the staff for outstanding teamwork, and enthusiastic participation in improving viral suppression for this group of complex urban patients.

## Author Contributions

**Conceptualization:** Debbie Y. Mohammed, Jihad Slim.

**Data curation:** Debbie Y. Mohammed.

**Formal analysis:** Debbie Y. Mohammed.

**Methodology:** Debbie Y. Mohammed, Jihad Slim.

**Resources:** Eugene Martin, Jihad Slim.

**Writing – original draft:** Debbie Y. Mohammed, Lisa Marie Koumoulos, Eugene Martin, Jihad Slim.

**Writing – review & editing:** Debbie Y. Mohammed, Lisa Marie Koumoulos, Eugene Martin, Jihad Slim.

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
