## [Decision Letter · Decision Letter 0]

22 Jul 2020

PONE-D-20-02167

Retrospective review of annual and durable retention in care and viral suppression, Peter Ho Clinic, 2013-2017

PLOS ONE

Dear Dr. Mohammed,

Thank you for submitting your manuscript to PLOS ONE. Please accept my personal apologies for the time it has taken to get a decision to you.After careful consideration, we feel that it has merit but does not fully meet PLOS ONE’s publication criteria as it currently stands.

Your manuscript has been evaluated by two external reviewers who have requested some clarifications about some aspects of your methodology, further discussion of key findings and further contextualisation with respect to existing literature (see the full reports below).

Therefore, we invite you to submit a revised version of the manuscript that addresses the points raised during the review process.

We look forward to receiving your revised manuscript.

Kind regards,

Dr Joseph Donlan

Associate Editor

PLOS ONE

Journal Requirements:

Additional Editor Comments (if provided):

Reviewers' comments:

Reviewer's Responses to Questions

**Comments to the Author**

1. Is the manuscript technically sound, and do the data support the conclusions?

Reviewer #1:Yes

Reviewer #2:Yes

2. Has the statistical analysis been performed appropriately and rigorously? 

Reviewer #1:Yes

Reviewer #2:Yes

3. Have the authors made all data underlying the findings in their manuscript fully available?

Reviewer #1:Yes

Reviewer #2:Yes

4. Is the manuscript presented in an intelligible fashion and written in standard English?

Reviewer #1:Yes

Reviewer #2:Yes

5. Review Comments to the Author

Reviewer #1:This is a well written and well articulated research work on an important topic in HIV management. However, the authors are invited to consider some clarification to further improve the understanding of the work.

Title: The authors could consider a review of the title to "Retention in care and viral suppression among HIV patients in Peter Ho Clinic"

Abstract: Line 36-37 What are the comparison group?

Method section:

The authors need to provide information on the study setting to help the readers understand the context of the clinic and its operational environment. There is need for more details about the patients management in the hospital and the total number of patients receiving HIV care in the hospital.

line 89-91 : How long had the newly diagnosed patients been on treatment? Were they excluded because they did not experience virologic failure or because they were not eligible for the test as at the the time the analysis were conducted? Could the authors provide more information about how many patients were excluded from the analysis and for what reasons in a flow diagram?

Line 98: Remove "The definition of"

line 98-100: In defining the annual and durable retention, what was the starting point? The start of study or the point the patient commenced on ART? There is need to clarify on this further for ease of reproducibility. It is also important because retention rate vary with the duration on care from commencement of ART. This is same for viral suppression.

line 109: How about 18 and 19 years of age?

line 121: How did you handle those that did not have insurance "none"? The constituted 21% of the total respondents.

line 129-132: Could the authors provide more details about the model development? How did they handle multiple collinearity? How were factors selected for the final model?

Results section:

Line 134: Were these all PLWH? Where they new enrollment within the period, all previously enrolled prior to this period or both new and old enrollments? This could have been explained in the method section for more clarity.

Table 2: Please check the figures, the total number of new in Table 1 is 157 and is 158 in Table 2. Also, the current was 1229 in Table 1 and 1230 in Table 2. Were there additions? If anything the numbers should be less if some died within the period. Kindly cross check the figures.

How was those without insurance handled. One would expect a footnote here explaining the merging of the cells in table 1.

line 160: "patient" status

Table 4:

The authors should consider rephrasing the title of table 4. I understand that these were the initial 935 patients that ever reported virologic failure whose latest results showed some changes probably after an adherence intervention and a follow up test. It should read factors affecting viral suppression among those that reported an earlier virologic failure at PHC 2013-2017

Discussion:

The authors should resist the urge of repeating the results in the discussion section. They should rather discuss the implication of their finding to practice and situated it in the body of knowledge. They can also provide unique context to help the readers understanding the findings and why they may defer from other well known findings.

Line 221:

The authors could do more search on longitudinal retention in care among PLWH in literature, what is described as durable retention here. Here some literature to consider

https://www.ncbi.nlm.nih.gov/pmc/articles/PMC5289300/

https://link.springer.com/article/10.1007/s10461-019-02450-7?shared-article-renderer

https://academic.oup.com/cid/article/62/5/648/2462795

Line 271: Was this a cohort study?

Reviewer #2:This paper reports a 14-percentage-point increase in HIV viral suppression over five years in the HIV-focused Peter Ho Clinic in urban New Jersey, without consistent increases to retention in care. These findings are inspiring, but the paper should frame them more compellingly, more clearly describe analytic decisions, and highlight and analyze their suppression initiative more.

Abstract

Please see comment below about conveying your unique angle and relevance throughout the paper, including in the abstract.

Introduction

Lines 52-54: Please make the statistics about viral suppression comparable in both sentences, i.e., both among (or not among) persons retained in care.

Why this set of years and why this clinic? What unique questions of broad relevance were you able to ask and answer? Share things that will help the reader see how this analysis informs not only where this leaves PHC patients and policies but also other populations and health care systems. For example, PHC is the first and largest HIV clinic in NJ, it is set within a large medical center that also provides social services, it accepts public insurance and is supported by federal Ryan White funding and many patients are low-income, NJ and Newark are racially and ethnically diverse and have a relatively high prevalence of HIV, etc. The particular angle and relevance of your paper should shine not only in the introduction but also the abstract and discussion.

Methods

Line 110: “Gender (male, female)” needs to be changed. Please specify if this is gender identity or sex assigned at birth. If the former, the terms “man” and “woman” are more appropriate, and it would also be helpful to state whether and how non-binary and transgender persons were ascertained and classified.

Your analyses used multiple outcomes, denominators, inclusion criteria, and time periods. This section needs to more clearly convey what exactly the analyses were and who was in each one. Places to insert expansions on this may be around lines 88-89 (“Patients included in this study were at least 18 years old and alive as of December 31st of the respective year.”); and lines 82-83 (“This study evaluated annual retention in medical care and viral suppression, durable retention in medical care and viral suppression among PLWH, in the PHC, from 2013 to 2017.”).

Lines 95-97 (“Annual retention in medical care included documentation of at least two medical

96 encounters at PHC in the respective year (Yes, No). A medical encounter was defined as a

97 medical visit with a prescribing provider or an HIV viral load test…”): Guidelines have been shifting to permit once-yearly testing for persons with durable suppression. Especially since you also have a durable retention measure, did you consider looking at retention as being at least once in the year? Or did clinic policy, prescribing practices, or the HIV/AIDS Bureau guidelines sway you to only look at twice-annual? See also my comments on the first paragraph of the Discussion.

Lines 98-100 (“The definition of durable retention in care was defined as having two medical encounters and ART in each calendar-year, over a two, three, four, or five-year period, from 2013 to 2017.”): The durable retention definition seems unclear. How is the length of the period determined? Might be more standard to just pick one and justify it based on the literature, federal guidance, or the interest and practices of your clinic. Later when I go to the figure, I sort of understand what you’ve done – you calculated as many of the durable measures as you could for each person according to their amount of… enrollment or follow-up time? – and it would be helpful to describe that in the methods.

Line 84 (“These results will be used to assess progress of the clinic towards achieving NHAS 2020 goals.”): Yes, every HIV clinic should be doing this internally. But again, what about your clinic, analysis, or findings makes this publishable because it’s of interest to a broader audience? Expand on this to tell the reader how the findings will illuminate broader truths about HIV care and suppression among attendees of an urban US HIV clinic.

Results

Tables 1 and 2: How come the patient status (current / new / reengaged) breakdown of the 1,000 persons retained in care in Table 2 seems inconsistent with that for the 1,000 2017 persons in Table 1? Are 705 persons current, or 908?

Lines 183-184 (“Viral suppression increased yearly: 2013 (73%), 2014 (76%), 2015 (80%), 2016 (81%) and 2017 (87%).”): What a substantial and exciting increase. Congratulations! You are almost at 90%. What changed to make this happen? Was it clinic practice, medication changes, the slight shift in patient load toward persons with fewer financial barriers to adherence? This could be the angle for your entire paper. With a patient population with as many challenges as yours re substance use, mental illness, and poverty, many would want to know how you worked with your patients to achieve an increase to 87% suppression.

Table 4: Are there any other characteristics about the care itself or contact with PHC or the suppression initiative that you’d want to include in a model? Or do you take the retention measure generally as a proxy for contact with the initiative? Could potentially break out retention differently to be more explicit about the nature of the care, e.g., instead of not retained vs. retained, could instead do something like not retained vs. retained but no case manager / nurse contact vs. retained and had a session with case manager / nurse.

Discussion

Lines 207-208 (“Annual retention in medical care was higher among PLWH, in 2017, in PHC compared to national reports, 90% and 71.1%, respectively”): Are these statistics comparable? Seems like the 71% nationally may be among all PLWH, whereas PHC patients are inherently connected with a provider and were in care during at least one point in the last five years.

Lines 211-220: Given your attention to findings about suppression among persons not meeting the HAB criteria for retention in care (i.e., who have <2 visits/year), why not also look at this, potentially with a single-visit measure of retention? As it is, this first paragraph feels a bit like it goes off on a tangent here. Maybe you could explore suppression among persons not retained, at this length, in later paragraphs and, in the first paragraph, highlight several of your most important findings, such as that you’re hitting 90-90-90 targets for retention and suppression, your five-year retention rates aren’t extraordinary, and you experienced substantial increases in suppression in the last five years.

Lines 227-236: Great job with your viral suppression initiative! Seems like a huge success. Why did you hold this key information until the Discussion, and bury it in the third paragraph? And have you published anything else about it that you can cite, too? Again, this is a big part of your story. Let the reader know in the abstract and the intro, methods, and discussion sections.

Line 236: I see that this was a data-driven initiative with multiple partners, which is terrific. Readers will also want to know what those “appropriate interventions” were and ideally a few measures of those interventions. For example, did referrals to drug treatment / mental health services / housing services / adherence counseling / peer support increase from 2013 (pre-initiative) to 2017, and can you enumerate that change? What else was done? Did other PHC characteristics change? I see that the patient load decreased from >1,200 to 1,000. Did funding or staffing change? Did the ratio of case managers to patients increase? All of this is your story. If you don’t tell it fully in this paper, I hope you’ll at least give us a little more info here, and write it up completely elsewhere, because it seems exciting.

Line 280 (“Improved viral suppression from 2013-2017 was possible due to involvement of all staff.”): Certainly. Did patients and/or your CAB also inform your suppression initiative? Do you want to acknowledge that in some way here or earlier?

Figure 4: Beautiful and encouraging. I’m not clear on how you’re able to calculate five-year suppression for 2014 persons – would’ve thought just 2013. Same for four-year suppression for 2015 persons, etc.

6. PLOS authors have the option to publish the peer review history of their article (what does this mean?). If published, this will include your full peer review and any attached files.

Reviewer #1:**Yes:**Chukwuma Umeokonkwo

Reviewer #2:No

While revising your submission, please upload your figure files to the Preflight Analysis and Conversion Engine (PACE) digital diagnostic tool,https://pacev2.apexcovantage.com/. PACE helps ensure that figures meet PLOS requirements. To use PACE, you must first register as a user. Registration is free. Then, login and navigate to the UPLOAD tab, where you will find detailed instructions on how to use the tool. If you encounter any issues or have any questions when using PACE, please email PLOS atfigures@plos.org. Please note that Supporting Information files do not need this step.

---

## [Author Response · Author response to Decision Letter 0]

13 Sep 2020

Reviewer #1: 

Title: The authors could consider a review of the title to "Retention in care and viral suppression among HIV patients in Peter Ho Clinic"

The title of the manuscript was revised as follows –Annual and durable HIV retention in care and viral suppression, Peter Ho Clinic, 2013-2017. 

Abstract: Line 36-37 What are the comparison group? 

Based on the word limit for the abstract, this sentence was deleted. 

Method section:

The authors need to provide information on the study setting to help the readers understand the context of the clinic and its operational environment. There is need for more details about the patient’s management in the hospital and the total number of patients receiving HIV care in the hospital.

The setting of the clinic was described as follows:

Setting. The PHC is located on the campus of Saint Michael’s Medical Center, an urban academic institution in Newark, New Jersey. Newark is the epicenter of the epidemic in New Jersey and lies in close proximity to New York City. This is the first clinic in the state to provide medical care for PLWH and serves approximately 1,200 persons yearly. Co-located services include HIV testing, access to pre-exposure prophylaxis and linkage to care coordinators. Clinical staff include infectious diseases providers and fellows, nurse practitioners, medical and non-medical case-managers, and a phlebotomist. Specialty practices include gynecology, medication assisted therapy for opioid use and pain management services. 

line 89-91: How long had the newly diagnosed patients been on treatment? Were they excluded because they did not experience virologic failure or because they were not eligible for the test as at the time the analysis were conducted? Could the authors provide more information about how many patients were excluded from the analysis and for what reasons in a flow diagram?

The sentence was rephrased to include the reason for exclusion: Six patients were excluded from the virologic failure analysis as their viral loads were less than 200 copies/mL at the time of diagnosis (Fig 1). 

Line 98: Remove "The definition of"

Done

line 98-100: In defining the annual and durable retention, what was the starting point? The start of study or the point the patient commenced on ART? There is need to clarify on this further for ease of reproducibility. It is also important because retention rate vary with the duration on care from commencement of ART. This is same for viral suppression.

I added the term – calendar year to the definitions of retention and viral suppression, for example, “Annual retention in medical care included documentation of at least two medical encounters, at least 90 days apart, at PHC in the respective calendar-year (Yes, No). “ 

Also clarified under study population as follows: 

Patients included in this study were at least 18 years old and alive as of December 31st of the respective year. At least one time in 2013-2017, they saw a medical provider, received a prescription for ART and had viral load results documented in the electronic medical record. They were included in the study at the time of their first medical visit or viral load to six months after the last documented viral load or medical visit

line 109: How about 18 and 19 years of age?

The age category was 18-29. 18 and 19-year-olds constitute a miniscule proportion of PLWH in PHC

line 121: How did you handle those that did not have insurance "none"? The constituted 21% of the total respondents. 

I added the following definition for patients without insurance: (the care of these patients was paid by charity care, state funding or Ryan White funding)

line 129-132: Could the authors provide more details about the model development? How did they handle multiple collinearity? How were factors selected for the final model?

I expanded on the data analysis section as follows: :

Multiple logistic regression models, using backwards deletion, were developed to identify factors associated with retention in medical care in 2017; virologic failure, 2013-2017; and viral suppression, 2013-2017, for those experiencing virologic failure. The selected variables were based on clinical and epidemiological significance. Collinearity in the categories for risk and drug use were mitigated by adhering to the hierarchical categorization developed by the Centers for Disease Control and Prevention [27], and categorizing drug use as cocaine only, heroin only and those with a history of using both drugs. Model fit was evaluated using Hosmer and Lemeshow Goodness-of-Fit test. 

Results section:

Line 134: Were these all PLWH? Yes – I deleted the word patients and added PLWH. 

Where they new enrollment within the period, all previously enrolled prior to this period or both new and old enrollments? This could have been explained in the method section for more clarity. 

We added the following definition under covariates, earlier in the manuscript: Patient’s status was defined as Current, New and Reengaged. Current patients were those in care in 2013. In the respective calendar-year, new patients were newly diagnosed and reengaged were PLWH who did not previously receive care in PHC.

Table 2: Please check the figures, the total number of new in Table 1 is 157 and is 158 in Table 2. Also, the current was 1229 in Table 1 and 1230 in Table 2. Were there additions? If anything the numbers should be less if some died within the period. Kindly cross check the figures.

Thank you for this comment. I reviewed all the tables, corrected the numbers, and included Figure 1. 

One would expect a footnote here explaining the merging of the cells in table 1. Done for each table.

line 160: "patient" status This change was made.

Table 4:

The authors should consider rephrasing the title of table 4. I understand that these were the initial 935 patients that ever reported virologic failure whose latest results showed some changes probably after an adherence intervention and a follow up test. It should read factors affecting viral suppression among those that reported an earlier virologic failure at PHC 2013-2017

The title of Table 4 was changed: Table 4: Factors Affecting Viral Suppression among patients with Virologic Failure, Peter Ho Clinic, 2013-2017

Discussion:

The authors should resist the urge of repeating the results in the discussion section. They should rather discuss the implication of their finding to practice and situated it in the body of knowledge. They can also provide unique context to help the readers understanding the findings and why they may defer from other well known findings.

Line 221:

The authors could do more search on longitudinal retention in care among PLWH in literature, what is described as durable retention here. Here some literature to consider

https://www.ncbi.nlm.nih.gov/pmc/articles/PMC5289300/

https://link.springer.com/article/10.1007/s10461-019-02450-7?shared-article-renderer

https://academic.oup.com/cid/article/62/5/648/2462795

Thank you for these references, we have reviewed and included them in the paper. The discussion section was revised extensively. 

Line 271: Was this a cohort study? Deleted the word “cohort”. Thanks

Reviewer #2: This paper reports a 14-percentage-point increase in HIV viral suppression over five years in the HIV-focused Peter Ho Clinic in urban New Jersey, without consistent increases to retention in care. These findings are inspiring, but the paper should frame them more compellingly, more clearly describe analytic decisions, and highlight and analyze their suppression initiative more.

Abstract

Please see comment below about conveying your unique angle and relevance throughout the paper, including in the abstract.

Introduction

Lines 52-54: Please make the statistics about viral suppression comparable in both sentences, i.e., both among (or not among) persons retained in care.

The sentence was changed: In comparison at the Peter Ho Clinic (PHC), of 1,229 PLWH in medical care in 2013, 84% were retained for one year, of whom 73% were suppressed.

Why this set of years and why this clinic? What unique questions of broad relevance were you able to ask and answer? I highlighted the following in the paper- 

This time-period is relevant as it is within the evaluation period of the NHAS, longitudinal evaluations are more relevant as we progress towards ending the epidemic. 

Share things that will help the reader see how this analysis informs not only where this leaves PHC patients and policies but also other populations and health care systems. For example, PHC is the first and largest HIV clinic in NJ, it is set within a large medical center that also provides social services, it accepts public insurance and is supported by federal Ryan White funding and many patients are low-income, NJ and Newark are racially and ethnically diverse and have a relatively high prevalence of HIV, etc. The particular angle and relevance of your paper should shine not only in the introduction but also the abstract and discussion.

We added an Introductory paragraph, in addition to expanding on the setting in the Methods section.

An estimated 1.04 million persons were living with HIV (PLWH) in the United States in 2018, with a prevalence of 374.6 per 100,000 population. [1] Males accounted for (75%), of whom 35% were Black, 73% were males who had sex with males (MSM), and injection drug use (IDU) was 9%. Among females, 58% were Black, of whom 77% reported heterosexual contact and 20% IDU as their transmission risk. In comparison, the prevalence rates in New Jersey and Essex County, were 419.7 and 1,194.9 per 100,000 population, respectively. [2] In Essex County, males accounted for 61% of PLWH, of whom 69% were Black, 35% were MSM, and 16% reported IDU. Among females 81% were Black, of whom 19% reported IDU and heterosexuals were 67%. The Peter Ho Clinic (PHC) is located in the City of Newark, in the county of Essex. The distribution of PLWH in Newark is like that of Essex County. [3]

Under methods – I included a description of the setting to contextualize the study. 

Setting. The PHC is located on the campus of Saint Michael’s Medical Center, an urban academic institution in Newark, New Jersey. Newark is the epicenter of the epidemic in New Jersey and lies in close proximity to New York City. This is the first clinic in the state to provide medical care for PLWH and serves approximately 1,200 persons yearly. Co-located services include HIV testing, access to pre-exposure prophylaxis and linkage to care coordinators. Clinical staff include infectious diseases providers and fellows, nurse practitioners, medical and non-medical case-managers, and a phlebotomist. Specialty practices include gynecology, medication assisted therapy for opioid use and pain management services. 

Methods

Line 110: “Gender (male, female)” needs to be changed. Please specify if this is gender identity or sex assigned at birth. If the former, the terms “man” and “woman” are more appropriate, and it would also be helpful to state whether and how non-binary and transgender persons were ascertained and classified. Restated as Sex at Birth

Your analyses used multiple outcomes, denominators, inclusion criteria, and time periods. This section needs to more clearly convey what exactly the analyses were and who was in each one. Places to insert expansions on this may be around lines 88-89 (“Patients included in this study were at least 18 years old and alive as of December 31st of the respective year.”); and lines 82-83 (“This study evaluated annual retention in medical care and viral suppression, durable retention in medical care and viral suppression among PLWH, in the PHC, from 2013 to 2017.”).

I expanded on both sentences as follows and added Figure 1: 

Population. Patients included in this retrospective observational study were at least 18 years of age and alive as of December 31st of the respective year. At least one time in 2013-2017, they saw a medical provider, received a prescription for antiretroviral therapy (ART) and had viral load results in the electronic medical record. Patients were eligible at the time of their first medical visit or viral load until six months after the last documented viral load or medical visit. Six patients were excluded from the virologic failure analysis as they did not have detectable viral loads at the time of diagnosis. 

This study evaluated annual and durable retention in medical care and viral suppression, among PLWH, in the PHC, from 2013 to 2017. These results will be used to assess progress of this clinic towards achieving NHAS 2020 goals. In addition, factors that may contribute or serve as barriers to achieving these goals will be identified to inform future interventions. 

A flow chart was added to describe the clarify the denominators used in different evaluations.

Lines 95-97 (“Annual retention in medical care included documentation of at least two medical encounters at PHC in the respective year (Yes, No). A medical encounter was defined as a medical visit with a prescribing provider or an HIV viral load test…”): Guidelines have been shifting to permit once-yearly testing for persons with durable suppression. Especially since you also have a durable retention measure, did you consider looking at retention as being at least once in the year? Or did clinic policy, prescribing practices, or the HIV/AIDS Bureau guidelines sway you to only look at twice-annual?

Current practice required by funders is that we have 2 medical visits and 2 viral loads each year. (goal).

I used the HIV/AIDS Bureau guidelines to arrive at the definitions of retention and durable retention as I think that this is more in line with reality which was described as 2 medical encounters – a medical visit or viral load. These are both considered as medical encounters – different than twice annual recommendations. 

Lines 98-100 (“The definition of durable retention in care was defined as having two medical encounters and ART in each calendar-year, over a two, three, four, or five-year period, from 2013 to 2017.”): The durable retention definition seems unclear. How is the length of the period determined? Might be more standard to just pick one and justify it based on the literature, federal guidance, or the interest and practices of your clinic. Later when I go to the figure, I sort of understand what you’ve done – you calculated as many of the durable measures as you could for each person according to their amount of… enrollment or follow-up time? – and it would be helpful to describe that in the methods.

I added the term calendar-year which I think would lend clarity- also the inclusion criteria as follows:

Durable retention in care was defined as having at least two medical encounters and ART in each calendar-year, over a two, three, four, or five-year period, from 2013 to 2017.

Patients were eligible from the time of their first medical visit or viral load until six months after the last documented viral load or medical visit.

Line 84 (“These results will be used to assess progress of the clinic towards achieving NHAS 2020 goals.”): Yes, every HIV clinic should be doing this internally. But again, what about your clinic, analysis, or findings makes this publishable because it’s of interest to a broader audience? Expand on this to tell the reader how the findings will illuminate broader truths about HIV care and suppression among attendees of an urban US HIV clinic.

The results of this study will be used to assess progress of this clinic towards achieving The National HIV/AIDS Strategy 2020 goals. In addition, factors that may facilitate or serve as barriers to achieving these goals, will be identified to inform future interventions and provide valuable lessons to other clinics, serving an urban population with similar challenges.

I also highlight the importance of longitudinal measures of retention and viral suppression. 

Results

Tables 1 and 2: How come the patient status (current / new / reengaged) breakdown of the 1,000 persons retained in care in Table 2 seems inconsistent with that for the 1,000 2017 persons in Table 1? Are 705 persons current, or 908?

Thank you for this comment. I reviewed the numbers. 

Lines 183-184 (“Viral suppression increased yearly: 2013 (73%), 2014 (76%), 2015 (80%), 2016 (81%) and 2017 (87%).”): What a substantial and exciting increase. Congratulations! You are almost at 90%. What changed to make this happen? Was it clinic practice, medication changes, the slight shift in patient load toward persons with fewer financial barriers to adherence? This could be the angle for your entire paper. With a patient population with as many challenges as yours re substance use, mental illness, and poverty, many would want to know how you worked with your patients to achieve an increase to 87% suppression.

We added the viral suppression intervention under Methods.

 Table 4: Are there any other characteristics about the care itself or contact with PHC or the suppression initiative that you’d want to include in a model? Or do you take the retention measure generally as a proxy for contact with the initiative? Could potentially break out retention differently to be more explicit about the nature of the care, e.g., instead of not retained vs. retained, could instead do something like not retained vs. retained but no case manager / nurse contact vs. retained and had a session with case manager / nurse.

Added the clinic-wide suppression intervention to the methods section and discussed the impact on increasing the viral suppression rate. 

Discussion

Lines 207-208 (“Annual retention in medical care was higher among PLWH, in 2017, in PHC compared to national reports, 90% and 71.1%, respectively”): Are these statistics comparable? Seems like the 71% nationally may be among all PLWH, whereas PHC patients are inherently connected with a provider and were in care during at least one point in the last five years.

Deleted this comment

Lines 211-220: Given your attention to findings about suppression among persons not meeting the HAB criteria for retention in care (i.e., who have <2 visits/year), why not also look at this, potentially with a single-visit measure of retention? As it is, this first paragraph feels a bit like it goes off on a tangent here. Maybe you could explore suppression among persons not retained, at this length, in later paragraphs and, in the first paragraph, highlight several of your most important findings, such as that you’re hitting 90-90-90 targets for retention and suppression, your five-year retention rates aren’t extraordinary, and you experienced substantial increases in suppression in the last five years.

Done in the discussion section 

Lines 227-236: Great job with your viral suppression initiative! Seems like a huge success. Why did you hold this key information until the Discussion, and bury it in the third paragraph? And have you published anything else about it that you can cite, too? Again, this is a big part of your story. Let the reader know in the abstract and the intro, methods, and discussion sections. 

This was moved to the Methods section

Viral Suppression Intervention. A clinic-wide viral suppression effort was implemented in 2014, involving all staff and targeting PLWH with viral loads of at least 200 copies/ml. Three teams were established, each composed of a clinician (either a nurse practitioner or doctor), a case manager, nurse, and operational staff. Viral load data were electronically downloaded, in rolling 1-year intervals, in two-month increments, from January 1, 2013 to December 31, 2017. Every two months, each team was provided with an updated list of patients with a virologic failure. A nurse practitioner reviewed genotype results and the electronic medical record to ensure that patients were on appropriate ART. Case managers and nurses assessed barriers to adherence and medical care and implemented appropriate interventions. 

Line 236: I see that this was a data-driven initiative with multiple partners, which is terrific. Readers will also want to know what those “appropriate interventions” were and ideally a few measures of those interventions. For example, did referrals to drug treatment / mental health services / housing services / adherence counseling / peer support increase from 2013 (pre-initiative) to 2017, and can you enumerate that change?

This would be difficult to enumerate as there were no data collected. Added the following to the Methods section

Patients with adherence issues came in weekly for prefilled medication boxes and were transitioned to prefilled pharmacy packages when they became suppressed. The registration staff were key in scheduling and doing phone reminders for appointments. Referrals for substance use and mental health were done for on and off-site locations. Patients with unstable housing were referred to the New Jersey Housing Collaborative for assistance. 

 What else was done? Did other PHC characteristics change? I see that the patient load decreased from >1,200 to 1,000. Did funding or staffing change? Did the ratio of case managers to patients increase? All of this is your story. If you don’t tell it fully in this paper, I hope you’ll at least give us a little more info here, and write it up completely elsewhere, because it seems exciting.

None of this happened. I think just the constant follow up by the staff made it happen. 

No measurable changes or significant funding increases occurred. By virtue of fluctuating number of patients there may have been subtle, transient, changes in case managers workloads, however no data was collected in this regard. I believe the consistency and diligence of follow up by the staff contributed significantly.

Line 280 (“Improved viral suppression from 2013-2017 was possible due to involvement of all staff.”): Certainly. Did patients and/or your CAB also inform your suppression initiative? Do you want to acknowledge that in some way here or earlier?

The CAB was not involved in developing the viral suppression initiative.

I added acknowledgements to both patients and staff in PHC 

Figure 4: Beautiful and encouraging. I’m not clear on how you’re able to calculate five-year suppression for 2014 persons – would’ve thought just 2013. Same for four-year suppression for 2015 persons, etc.

Used data up to December 2019 – added to Methods section

---

## [Editor Report · Decision Letter 1]

16 Oct 2020

PONE-D-20-02167R1

Annual and durable HIV retention in care and viral suppression among patients of Peter Ho Clinic, 2013-2017

PLOS ONE

Dear Dr. Mohammed,

Thank you for submitting your manuscript to PLOS ONE. After careful consideration, we feel that it has merit but does not fully meet PLOS ONE’s publication criteria as it currently stands. Therefore, we invite you to submit a revised version of the manuscript that addresses the points raised during the review process.

Thank you for your substantial revisions to the paper. This looks much better. The descriptions of the clinic, its epidemiology vs the US HIV epidemic, and its initiative for unsuppressed PLWH are more thorough. I have a few additional suggestions, with line numbers from the manuscript version with changes tracked.

Line 180-181, "From 2009-2016, durable viral suppression was 37% in New York City, for the two and three-year period": Please provide two percentages here if reporting suppression over two and three years.

Line 216-218, "They were included in the study at the time of their first medical visit or viral load to six months after the last documented viral load or medical visit": It's the first visit/VL in the calendar years you looked at, rather than the first for the patient after diagnosis or coming to the clinic, right? Would be clearer to say, for example, "... of their first medical visit or viral load in 2013-2017."

Line 224 and throughout: The term "intervention" can suggest a set of actions that are limited in time or scope. Did you basically permanently restructure clinic services for all unsuppressed PLWH, with no comparison group? May want to use a term other than "intervention" or clarify that this intervention basically became the new way you're delivering services.

Line 300, "...categories of the following covariates were collapsed... insurance (Public, Private)": Please specify which group included the many uninsured patients.

Line 337, "Patients’ status included current patients 1229 (70%), 157 (9%) newly diagnosed and 373 (21%) who reengaged in medical care": Please revise so wording is less awkward, e.g., could say "By patient status, this included 1229 current patients (70%), 157 (9%) newly diagnosed patients, and 373 (21%) patients who reengaged in medical care."

Line 501-503, "This increase is attributed to the implementation of a clinic-wide viral suppression effort, involving all staff starting in 2014": Without a comparison to other PLWH in your clinic, Newark/Essex County, or the US, where VS may have also increased over these years, it would be more appropriate to swap "is attributed to" for "coincides with" to avoid suggesting causality.

Line 586-612, "In this study, PLWH in public housing were...": The consideration of housing and insurance is important. Please tie the literature back to your findings more. For example, how do you resolve the findings that PHC's public housing residents did worse but housing in general is known to be helpful? And the connection between insurance challenges and retention is very plausible but would be even stronger if you could provide evidence that PLWH at PHC had challenges reenrolling in Medicaid that affected their retention in care, and if you could square that with the fact that PHC also (according to your tables) sees uninsured persons.

Hopefully all of these can be addressed before publication, butrewording "intervention" is optional.

If applicable, we recommend that you deposit your laboratory protocols in protocols.io to enhance the reproducibility of your results. Protocols.io assigns your protocol its own identifier (DOI) so that it can be cited independently in the future. For instructions see:http://journals.plos.org/plosone/s/submission-guidelines#loc-laboratory-protocols

We look forward to receiving your revised manuscript.

Kind regards,

Ellen Wiewel

Academic Editor

PLOS ONE

Additional Editor Comments (if provided): see above

While revising your submission, please upload your figure files to the Preflight Analysis and Conversion Engine (PACE) digital diagnostic tool,https://pacev2.apexcovantage.com/. PACE helps ensure that figures meet PLOS requirements. To use PACE, you must first register as a user. Registration is free. Then, login and navigate to the UPLOAD tab, where you will find detailed instructions on how to use the tool. If you encounter any issues or have any questions when using PACE, please email PLOS atfigures@plos.org. Please note that Supporting Information files do not need this step.

---

## [Author Response · Author response to Decision Letter 1]

28 Nov 2020

Dear Editor and Reviewers, 

Thank you for your comments. 

Please find the revisions outlined below 

Line 180-181, "From 2009-2016, durable viral suppression was 37% in New York City, for the two and three-year period": Please provide two percentages here if reporting suppression over two and three years. 

This sentence was clarified as follows: 

In a study evaluating the impact of a care coordination program, from 2009-2016, durable viral suppression was 37% in New York City, in months 13-36 of follow-up. 

Line 216-218, "They were included in the study at the time of their first medical visit or viral load to six months after the last documented viral load or medical visit": It's the first visit/VL in the calendar years you looked at, rather than the first for the patient after diagnosis or coming to the clinic, right? Would be clearer to say, for example, "... of their first medical visit or viral load in 2013-2017." 

This sentence was clarified as follows: 

They were included in the study at the time of their first medical visit or viral load in 2013-2017 to six months after the last documented viral load or medical visit. 

Line 224 and throughout: The term "intervention" can suggest a set of actions that are limited in time or scope. Did you basically permanently restructure clinic services for all unsuppressed PLWH, with no comparison group? May want to use a term other than "intervention" or clarify that this intervention basically became the new way you're delivering services. 

This sentence was changed as follows: 

Beginning in 2014, services were restructured for patients with viral loads of at least 200 copies/mL. 

Line 300, "...categories of the following covariates were collapsed... insurance (Public, Private)": Please specify which group included the many uninsured patients. 

The definitions were included as follows: 

insurance (Medicaid, Medicare, private, none [the medical care of these patients were paid by charity care, grant funding from the state or Ryan White HIV/AIDS Program]) 

Further: 

In the regression analyses, categories of the following covariates were collapsed where the outcomes were similar: age (<60, > 60 years), race (non-Hispanic Black or Hispanic, Other), insurance (Public[included Medicaid, Medicare, None], Private) and drug use, (Yes, No). 

Line 337, "Patients’ status included current patients 1229 (70%), 157 (9%) newly diagnosed and 373 (21%) who reengaged in medical care": Please revise so wording is less awkward, e.g., could say "By patient status, this included 1229 current patients (70%), 157 (9%) newly diagnosed patients, and 373 (21%) patients who reengaged in medical care." 

This sentence was revised as suggested. 

Line 501-503, "This increase is attributed to the implementation of a clinic-wide viral suppression effort, involving all staff starting in 2014": Without a comparison to other PLWH in your clinic, Newark/Essex County, or the US, where VS may have also increased over these years, it would be more appropriate to swap "is attributed to" for "coincides with" to avoid suggesting causality. 

This sentence was modified as follows: 

This increase coincides with the restructuring of clinic services for patients with a viral load at least 200 copies/ml, involving all staff starting in 2014. 

Line 586-612, "In this study, PLWH in public housing were...": The consideration of housing and insurance is important. Please tie the literature back to your findings more. For example, how do you resolve the findings that PHC's public housing residents did worse but housing in general is known to be helpful? 

This paragraph was clarified as follows: 

In this study, patients who lived in public compared to private housing were less likely to achieve viral suppression. They included those who were unstably housed. Among patients accessing care in Ryan White funded sites in 2018, those who lived in unstable housing had lower rates of viral suppression compared to those who lived in stable housing (72.4% versus 88.4%), respectively. [37] Previous reports indicate that public housing mitigate barriers related to viral suppression. A recent retrospective matched cohort study in New York City reported that low income PLWH who received housing services for more than one year, were more likely to be engaged in care and virally suppressed than similar populations who did not receive these services [38]. In another New York City study, PLWH in continuous, stable housing in 2015, were reported to have continuous, stable viral suppression. [38]. Conversely, time spent in emergency housing was a predictor of a lack of viral suppression. Identifying patients in unstable housing and transitioning them to a stable environment will improve viral suppression rates in this subset of patients. 

I also identified another limitation 

Fifth, in this study we evaluated public and private housing only. In future evaluations, also classifying housing as stable or unstable may improve the granularity of the results. 

And the connection between insurance challenges and retention is very plausible but would be even stronger if you could provide evidence that PLWH at PHC had challenges reenrolling in Medicaid that affected their retention in care, and if you could square that with the fact that PHC also (according to your tables) sees uninsured persons. 

Changes were made to the discussion paragraph re insurance 

Insurance status was another structural barrier identified in this study. New Jersey expanded Medicaid and increased access to care for low-income residents, by implementinged in accordance with many reforms recommended by the Affordable Care Act starting in 2014  [4038]. Despite the availability of public insurance, before being seen in PHC, there were many processes patients had to navigate to make this a reality., These includeding qualifying for hospital charity care if they did not have insurance [41], enrollment and reenrollment, havingand  requiring a primary care provider to issue and referrals if they had a managed care plan,  before being seen in PHC. This may have contributed to patients not being retained in medical care or achieving viral suppression. Patients who are lost to care, more symptomatic, and/or racial/ethnic minorities may be less likely to overcome these barriers [4239]. At least 70% of PLWH at PHC, qualified for federally funded services, based on an income less than the FPL, for one person, in 2017, or had either Medicaid or no insurance (Charity Care/Ryan White funding), and were primarily racial and ethnic minorities (>90%). Numerous studies have documented the importance of Ryan White federal funding in maintaining PLWH in care and improved outcomes [30, 40-44-47]. Urban clinics will need to maximize the use of this resource to ensure that patients overcome structural barriers and achieve the goals of the NHAS.

---

## [Editor Report · Decision Letter 2]

9 Dec 2020

Annual and durable HIV retention in care and viral suppression among patients of Peter Ho Clinic, 2013-2017

PONE-D-20-02167R2

Dear Dr. Mohammed,

We’re pleased to inform you that your manuscript has been judged scientifically suitable for publication and will be formally accepted for publication once it meets all outstanding technical requirements.

Kind regards,

Ellen Wiewel

Guest Editor

PLOS ONE

Additional Editor Comments (optional):

Thank you for your revisions. Congratulations.
---

## [Editor Report · Acceptance letter]

14 Dec 2020

PONE-D-20-02167R2 

Annual and durable HIV retention in care and viral suppression among patients of Peter Ho Clinic, 2013-2017 

Dear Dr. Mohammed:

I'm pleased to inform you that your manuscript has been deemed suitable for publication in PLOS ONE. Congratulations! Your manuscript is now with our production department. 

Kind regards, 

on behalf of

Dr. Ellen Wiewel 

Guest Editor

PLOS ONE